# Investigation of an acute neurological outbreak in Eluru, India, 2020

Mahesh Kumar Mummadi[1], Raghavendra Pandurangi[1], J. J. Babu Geddam[1], Sukesh Narayan Sinha[1]*, Ananthan Rajendran[1], Sivaperumal P.[2], Naveen K. Ramachandrappa[3], Sree Ramakrishna K.[3], Pagidoju Sreenu[3]

1 Scientist, National Institute of Nutrition (ICMR-NIN), Indian Council of Medical Research, Hyderabad, India, 2 National Institute of Occupational Health (ICMR-NIOH), Indian Council of Medical Research, Ahmedabad, India, 3 Technical Officer, National Institute of Nutrition (ICMR-NIN), Indian Council of Medical Research, Hyderabad, India

☺ These authors contributed equally to this work.
* sukeshnr_sinha@yahoo.com

**Data Availability Statement:** All relevant data are within the manuscript and its Supporting information files.

## Abstract

On 4th December 2020, a sudden outbreak, with neurological symptoms like seizures, loss of consciousness etc., was reported in a town from south India. By 3rd day about 400 people were involved. A multi disciplinary team from our institute visited the site to investigate the outbreak. Based on the case history and clinical examination of the patients, the team suspected a probable diagnosis of an acute pesticide, heavy metal and/or mycotoxin exposure for which, biological samples (blood, urine) were collected from those who reported the symptoms as well as from a few who did not report symptoms (controls). To identify the source, water and food samples were collected. The samples were subjected to ICP-MS for heavy metal analysis, LC-MS/MS for pesticide analysis, microbiological analysis and ELISA-Kit method for aflatoxins if any. Clinical and dietary details were collected from a total of 112 participants, of which, 103 cases (77 active cases at Hospital and 26 recovered cases from community) and 9 were controls. A total of 109 biological samples, 36 water samples and food samples were collected. The mean age of the study participants was 29.2 years. Among cases, Seizures were seen in 84%, loss of consciousness in 66%, mental confusion in 35%, pinpoint pupil in 11%. Triazophos (organophosphate) pesticide was present in 74% of Blood samples and its metabolites were present in 98% of the urine samples collected from the cases. All the ten heavy metals investigated including lead, mercury and nickel were found to be within permissible limits except for a few samples. No presence of mycotoxins was observed in Food samples. Water samples which included Head pump and reservoir were free from pesticides; however, all water samples from households of cases had triazophos pesticide with a mean concentration of 1.00 ug/L. Thus, it was concluded that, the probable cause of outbreak was Triazophos (Organophosphate) pesticide contamination in water at the Household level. Regular surveillance for the presence of residual pesticides in soil, water and food with heightened vigour is recommended to prevent future outbreaks.

**Funding:** Indian Council of Medical Research is the premier agency to conduct medical research in the country, of which National Institute of Nutrition is a subsidiary. The institute was requested to conduct the epidemiological investigation by the government of Andhra Pradesh. Study is funded from internal funds of ICMR-NIN. Funding department of NIN has not involved in data collection and analysing and report preparation.

**Competing interests:** The authors have declared that no competing interests exist.

## 1. Introduction

A Sudden increase in occurences of a disease in a particular time and place is referred as Outbreak. It could be due to an infective or environmental (water or food contamination) origin. Foodborne illness is a serious public health threat as Food can become contaminated at any point from the farm-to-table continuum [1]. As per the analysis of about 4093 reported foodborne outbreaks, almost 70% were attributable to Salmonella, Norovirus and E. coli contamination [2], the remaining 30% were due to other various causes. One such contaminant is Pesticide use (intentional (suicide) [3] or unintentional [4]), which led to global outbreaks occurred in Glamorgan, UK (1956), Doha, Qatar (1967), Hofuf, Saudi Arabia (1967), Chiquinquira, Colombia (1967), Tijuana, Mexico (1967), Pasto, Colombia (1977), and Taucamarca, Peru (1999) due to pesticide contamination [5]. In 2008, Orissa, India, two deaths and 65 casualties happened due to damaged drinking pipes contaminating with Folidol, an organophosphate pesticide. Affected were having neurological symptoms like seizures, vomitings and myosis [6]. Similarly in 2013, 23 children were died and more than 48 required treatment due to monocrotophos, an organophosphate pesticide contamination with school lunch in Bihar, India [5].

On 4th December 2020, A sudden outbreak was reported with neurological symptoms like seizures, history of loss of consciousness, drowsiness, altered sensorium, frothing from the mouth, among some individuals in Eluru town of Andhra Pradesh state of India. No outbreaks of similar nature were reported in the study area in the recent past. A multidisciplinary team from our institute consisting of Clinical Epidemiologists, a Nutritionist, a Food Chemistry Expert, a Toxicologist, a Microbiologist and an Anthropologist visited the site to comprehensively investigate the outbreak. As on 7th December, 418 cases and a fatality were reported. The symptoms were initially presented among children below 8 years of age but gradually appeared in all age groups. Cases were admitted and treated at the Local Government Hospital with a provisional unknown neurological diagnosis. The outbreak had not only created panic among the residents but also attracted the attention of the national and international media as a mystery illness [7–10].

The clinical picture in almost all of the cases were consistent and had a similar pattern, there was a sudden onset of drowsiness, followed by seizures of mostly, generalized tonic-clonic type followed by fall and loss of consciousness for 3–30 minutes. Since the onset of drowsiness, cases did not remember anything after recovery. Due to falls and seizures, a few had sustained injuries like tongue bites, head and limb wounds. Altered sensorium (Mental confusion) was presented in many cases and very few had nausea and vomiting episodes during or after the seizures. Conspicuously, there was an absence of fever and diarrhoea. Pupillary reactions were sluggish in a small proportion of patients. Characteristically, the recovery from the episode was quick and complete and no residual neurological symptoms were reported. No patient was comatose.

With this background in mind and having no distinctive pointers towards the diagnosis, the team considered all possible potential causes and did a comprehensive exploratory analysis with biochemical, toxicological, mycological, microbiological, dietary and epidemiological perspective with an objective to examine the presentation of cases to identify probable causes for the outbreak and to propose measures to control the outbreak which can prevent further occurrences.

## 2. Materials and methods

### 2.1 Study design

The present epidemiological investigation was conducted as a case-control study, with controls taken from healthy neighbours of the cases. A quick clinical examination of the cases admitted

to the hospital was done with an aim to narrow down to three to four differential/probable diagnoses. Based on this, appropriate samples for the necessary investigation were collected. These samples included biological samples for confirming the presence of a causative agent and its detectable form in the body as well as in the suspected source. The presence of the same causative agent in these suspected sources was essential to form the corroborative evidence in clinching the final diagnosis.

## 2.2 Case definition

The case was defined as "any individual with sudden onset of neurological symptoms including but not limited to seizures, loss of consciousness, drowsiness, altered sensorium, frothing from the mouth". Patients with acute febrile seizures and with a past history of epilepsy were excluded.

## 2.3 Differential diagnosis

From the epidemic curve (Fig 1) which shows a sudden onset, steep rise, peaking and subsequent fall in the number of cases reported as on 9th December 2020, it was expected as single episode outbreak. Absence of fever indicated unlikelihood of diseases of infectious origin and contamination of unknown origin was the primary suspect. From the clinical picture, the following differential diagnoses were arrived at–(i) Acute Heavy Metal Contamination; (ii) Acute pesticide Contamination [Organo phosphate (OP)/ organo chlorine compounds (OC)] or (iii) Mycotoxin exposure–based on the following reasons as described in Table 1.

For the observed distribution of the cases within a short duration of time, the suspected common source(s) of contamination were drinking water, milk of the same packaging, cereals or pulses from a common distribution point, vegetables of the same produce, i.e., distributed from the same market and/or common food source consumed at a feast (which was not evident). Hence, the samples of drinking water at various levels of distribution, milk, rice/dal samples, vegetables from households and market, were collected and analysed.

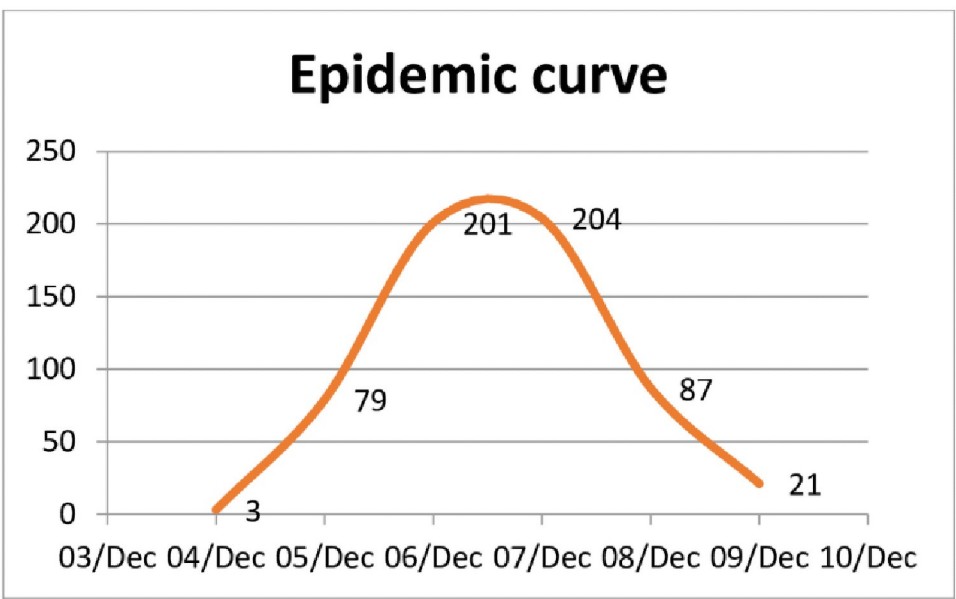

**Fig 1. Epidemic curve of the reported cases during the outbreak.**

**Table 1. Differential diagnosis.**

| Suspected contamination | Points in favour | Points against |
|---|---|---|
| Acute Lead contamination | • The clinical picture of neurological symptoms<br>• Possibility of a subclinical presentation explaining symptoms among selective family members | • Absence of GI symptoms<br>• Short-lived symptoms |
| Organo phosphate contamination | • CNS findings characteristic of OP contamination<br>• Could be a pesticide seepage into water bodies<br>• A few cases (10.7%) showing pinpoint pupils | • Absence of pronounced Muscarinic and Nicotinic features<br>• Non-essentiality of Atropine in therapy<br>• Severity not approximating to the fatalities |
| Mycotoxin exposure | • The clinical picture of neurological symptoms | • Absence of other features (dystonia, spasms, etc)<br>• Fatality rates go up to 10%, not in line with the outbreak |

### 2.4 Community sampling

Each municipal ward was considered as a cluster. Of the clusters which had reported cases, four were selected randomly for sample collection at community–JP Colony, Southern street, Chodidibba, Pension Mohalla. One cluster (Tangellamudi) which had no cases reported till the date of the investigation was included to serve as a control group. In the clusters with cases, few neighbours of the cases not affected were included as controls.

### 2.5 Investigations conducted

- Clinical history was collected and examination was conducted on patients undergoing treatment at the Government Hospital and at their respective homes for those who recovered. Dietary history was collected from these cases and additionally from some controls in the community.

- Biological samples: venous blood (5ml) and urine samples (15 ml) were collected under strict aseptic conditions from acute cases at the hospital and from recovered cases and controls at the community. Blood samples were collected in Vacutainer tubes and urine samples in small sterile plastic containers. Aliquots are prepared and transported under 4° C. Later analysed for presence of contaminant.

- Food samples: Raw samples of common food consumables like rice, dal, vegetables, milk and oil were collected from the households of the cases and controls from the community.

- Water samples: water samples at various levels like raw water before filtration at water head pump, chlorinated water after filtration, water from overhead reservoirs, and water at the consumption points in the community were collected in both cases and controls households. A few water samples from supplies on 4th and 5th December (during the initial days of the outbreak) have also been collected at the household level. water samples were collected in coded sterile containers from each Household and transported to the laboratory in an ice box which maintained < 4° C temperature. These samples were processed immediately after reaching laboratory.

- A semi-structured interview with the Executive Engineer of waterworks head pump at Eluru was conducted for gaining insights into the water supply mechanism of the town and for eliciting information regarding any unusual occurrences. The similar interview was also conducted with the dump yard handler to explore if there was any large quantity dumping of contaminants.

## 2.6 The methodology of analysis of heavy metals, pesticides, mycotoxins and microbiological Toxicities

**2.6.1. Heavy metals analysis.** The samples were analysed using Inductively Coupled Plasma Mass Spectrometry (ICP-MS) (AOAC 2013.06) method [11] for heavy metals like lithium, chromium, cobalt, nickel, arsenic, selenium, molybdenum, cadmium, antimony and mercury, whereas lead was analysed using ELAN 9000 (Perkin Elmer SCIEX).

**2.6.2. Pesticide analysis.** *Sample preparation.* A blood sample of 100 μl volume was taken in a 10 ml glass tube, and 900 μl of ACN was added. The mixture was vortexed for 10 minutes and passed through 0.2 μn syringe filter column, and the resulted eluant was collected in a clear 1 ml capacity vial and analysed on Liquid Chromatography -Mass spectrometry (LC–MS/MS).

*Concentration and mass spectrometry analysis.* The concentration of the large pesticides were analyzed on a 400 Qtrap triple, quadruple hybrid mass spectrometer of the applied biosystem, foster city CA, USA) make in accordance with the previous report method [12–14]. The analytes were investigated on multiple reaction positive electron spray ionization mode (ESI) and high-resolution monitoring. A high-end ultra-speed liquid chromatography powered with a triple. Quadrupole ion trap mass detector was used in multiple reaction monitoring mode. A batch comprising five aliquots of blood sample was taken for quantification, as well as a confirmation the percentage of each pesticide recovery was calculated against the known concentration with the analyzed concentration, in compliance with the previous reported [12–14].

Sigma Aldrich standards for blood samples with Detection limit- 0.037 nano gram/ml. The percentage recovery of Triazophas in blood ranged from 95 to 99% at concentrations of 1,5,10,100 & 200 ug/L and CDC Atlanta standards were used for urine samples with detection limit- DMP—0.058 nano gram/ml & DEDTP—0.0198 ug/L[DAP (Di-alkyl Phosphate) = DMP (Dimethyl Phosphate) +DEDTP (DiEthyl Di-Thio phosphate)]' for water samples with detection limit- 0.028 ug/L, and vegetables samples with detection limit-0.087 nano gram/gram was used [12–14].

**2.6.3. Mycotoxin estimation.** Total-Aflatoxin ELISA kit method (Pribo Fast EKT-011-48/96T Kit) was used.

**2.6.4. Microbiological: Analysis.** The standards set by the Bureau of Indian Standards (Drinking water specification- Second Revision of IS 10500) were considered [15].

## 2.7 Ethical approval

Ethical Clearance was obtained from the Indian Council of Medical Research- National Institute of Nutrition Institutional Ethical Committee. **IEC Registration Number- ECR/35/Inst/AP/2013, Study Protocol Number-1/I/2021**. Informed written consent was taken from the study participants to obtain the samples after explaining the procedures. Necessary approvals were sought from the respective departments in the Government to carry out the study.

**2.7.1 Permission to conduct the study.** On December 6th 2020, Director, ICMR- NIN (Indian Council of Medical Research-National Institute of Nutrition, was requested by Commissioner, Ministry of Health and Family welfare,Andhra Pradesh State, India to investigate into the Acute Neurological outbreak.

Secretary, Department of Health Research, Ministry of Health & Family Welfare, Government of India and Director General, Indian Council of Medical Research also consented to attend the epidemic.

## 2.8 Statistical analysis

Data were entered into MS Excel (Microsoft,USA) and analysed using Epi Info 7.2 version (CDC,USA). (Supporting information) Percentages were calculated. ANOVA test was performed for blood concentrations of Triazophos pesticide in cases which are collected within 24, 48 and 72 hours.

## 3. Results

Clinical and dietary details were collected from a total of 112 participants. Out of which, 103 cases (77 active cases admitted at District Hospital and 26 recovered cases from the community) and 9 were controls. A total of 109 biological (blood and urine) samples, 36 water samples and other food samples were collected.

Affected individuals were more commonly males than females and mostly aged below 40 years with a mean age of 29.2 years(Fig 2a). Seizures and loss of consciousness were the most common clinical presentations followed by frothing, altered sensorium, nausea and vomiting. Pinpoint pupil was present in 10.7% cases examined and was mostly among those presenting with the onset of symptoms on the same day or the day before the examination (Fig 2b). Except for 2–3 relapse cases, most of the patients who were discharged from the hospital did not show recurrence of symptoms.

### 3.1 Diet consumption patterns

Almost all the patients admitted in the hospital, the cases and the controls in the affected community were using piped drinking water supplied by the local municipality and the water source was common. Few of the patients had the habit of boiling water and filtering it before consumption. A consistent and common pattern of using municipal water for drinking was found among all the cases, while in the control areas, 6 out of 9 Households purchased bottled water regularly. A significant finding reported by the households was that the municipal water supply was muddy and coloured with bad odour on days preceding the outbreak. Most families were consuming rice and dal procured from the public distribution system. Tomato was the most frequently consumed vegetable. Most of the households were purchasing unpackaged milk from different local vendors.

### 3.2 Biochemical analysis of samples for pesticides

A total of 90 blood samples and 51 urine samples of the cases, 11 blood samples of controls, 24 samples of water at various levels of supply chain and 10 samples each of tomatoes and brinjals were analysed for the presence of pesticides.

The blood and urine samples were tested for 37 different types of pesticides. Triazophos (organophosphate) was present in 67 out of 90 (74%) blood samples of cases. Fig 3 shows that the pesticide concentrations in blood were higher in cases where samples were collected within 24 hours of the onset of symptoms (mean concentration-17.6 ug/L) than those samples that were collected later than 24 hours (mean concentration- 8.01 ug/L) or 48 hours(mean concentration-5.7 ug/L). The difference, however, was not statistically significant (ANOVA f-ratio = 1.585; p-value = 0.213). Almost all urine samples (50 out of 51) analysed have shown the presence of Organophosphates (Dialkyl phosphate with a mean concentration of 0.92 ug/L.

The water samples (26 Nos) were analysed for the presence of triazophos,of which, 13 were from cases households, 7 from controls households, 3 samples of stored water supplied on 4th and 5th December (during the outbreak) and 1 sample from common source reservoir at JP Colony. 2 samples from the head pump before and after filtration. Head pump and reservoir

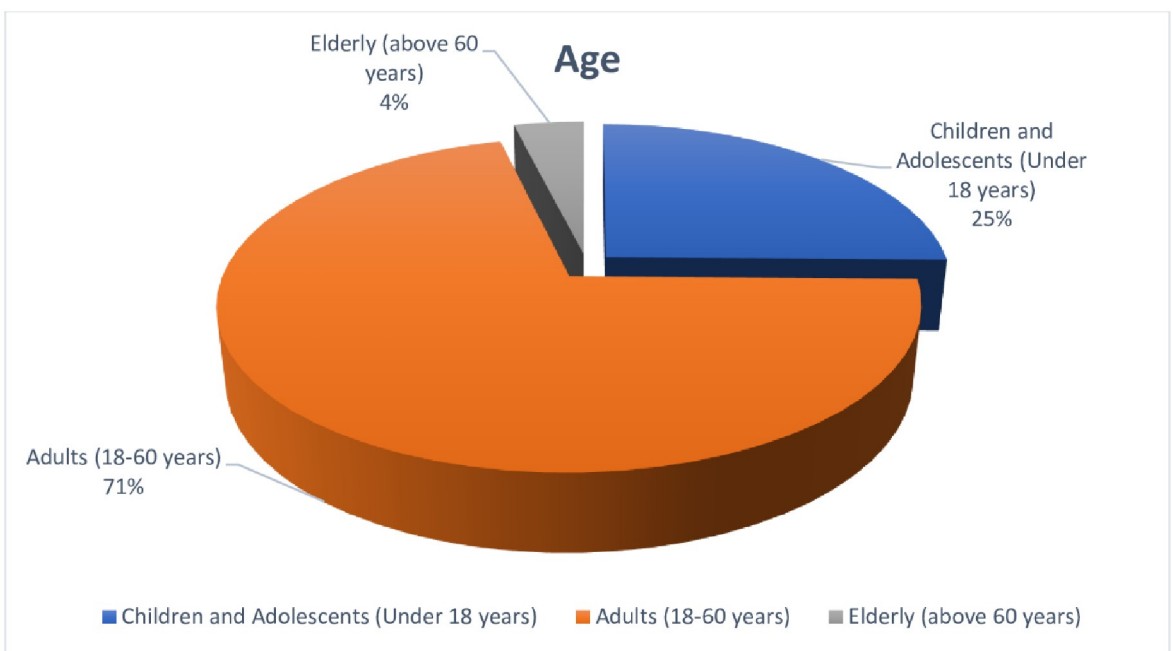

a

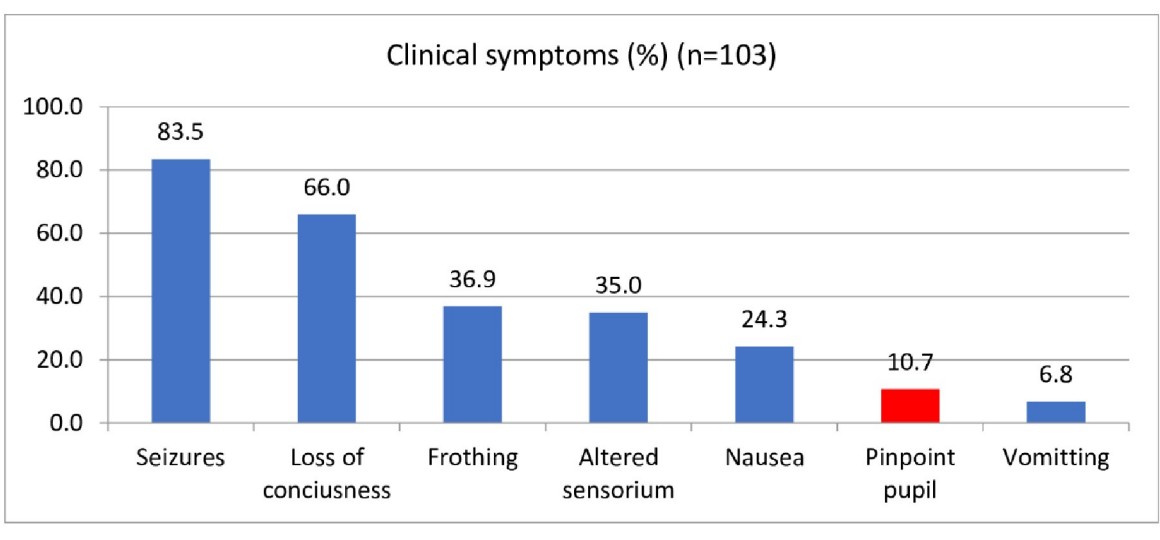

b

**Fig 2.** a. Age distribution of all cases (Hospital and community). b. Distribution of Clinical Symptoms in cases admitted in Local Government hospital.

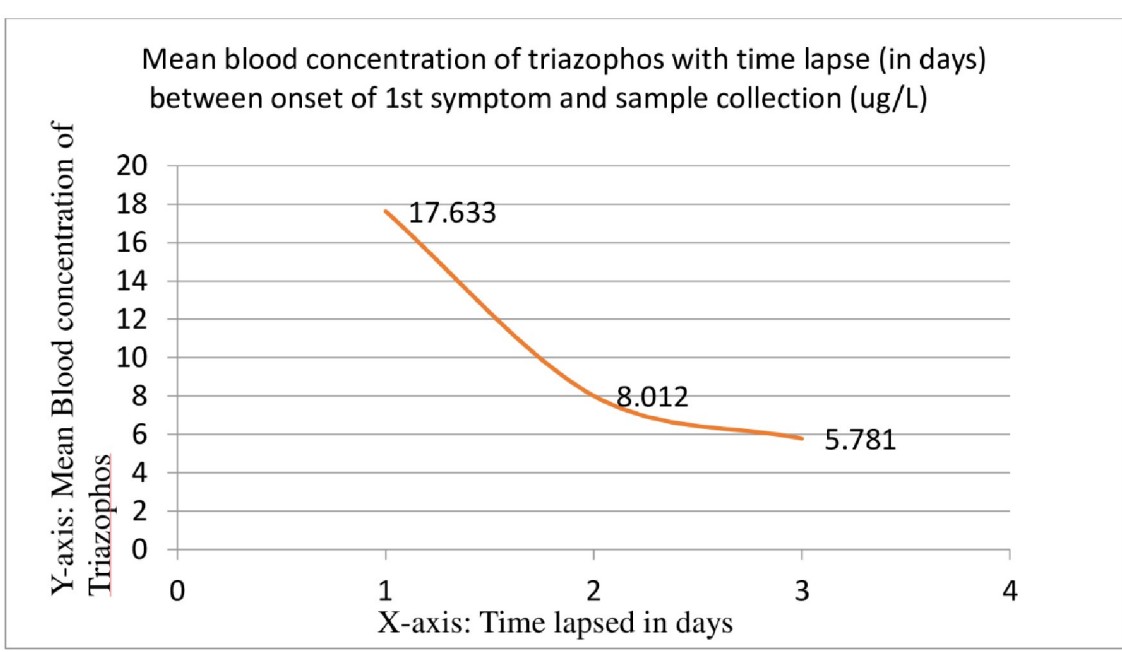

**Fig 3. Mean Blood concentration of Organo Phosphate Pesticides (Triazophos) over the period of onset of first symptoms to blood sample collection.**

samples are free from pesticides. Almost All water samples from households of cases had the presence of pesticides (Triazophos) with a mean concentration of 1.00 ug/L (Table 2).

Vegetable samples: Ten samples each of tomatoes and Brinjals were analysed. It was observed that Metribuzin, a herbicide,was present in all tomato and brinjal samples (100%). Mean value of Metribuzin in Tomatoes was 61.3 ng/gram and brinjals was 6.72 ng/gram.

Fig 4a–4c show the chromatogram of Triazophas in standard, blood and water samples respectively.

**3.2.1 External quality assurance for pesticides.** The Blood and water samples were sent to the Indian Council of Medical Research–National Institute of Occupational Health (ICMR-NIOH), Ahmedabad for external quality assurance. In ICMR-NIOH, the samples were analysed on High-resolution synapte mode LC-MS/MS and the Presence of Triazophas in blood and water was confirmed.

## 3.3 Biochemical analysis of samples for heavy metals

Heavy metals like lithium, chromium, cobalt, nickel, arsenic, selenium, molybdenum, cadmium, antimony, mercury and lead were analysed in blood, urine, water, vegetables, rice, dal and milk samples. The lead was found above the acceptable limits in 2 out of 102 blood

**Table 2. Percentage of pesticide identified in blood, urine and household water samples of cases and controls.**

| S. No. | Type of Sample | Cases | | | Controls | | |
|---|---|---|---|---|---|---|---|
| | | Number of samples analysed | Samples above permissible limits | Mean concentration | Number of samples analysed | Samples above permissible limits | Mean concentration |
| 1 | Blood | 90 | 67 (74%) | 10.48 ng/ml | 11 | 1(9%) | 2.82 ng/ml |
| 2 | Urine | 51 | 50 (98%) | 0.92 ug/L | Nil | Nil | Nil |
| 3 | Water | 13 | 12 (92%) | 1.00 ug/L | 7 | 1(14%) | 0.08ug/L |

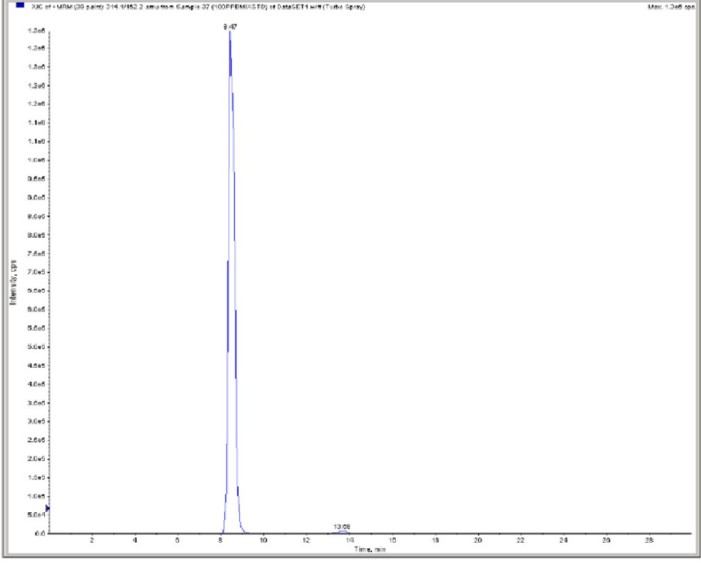

**a**

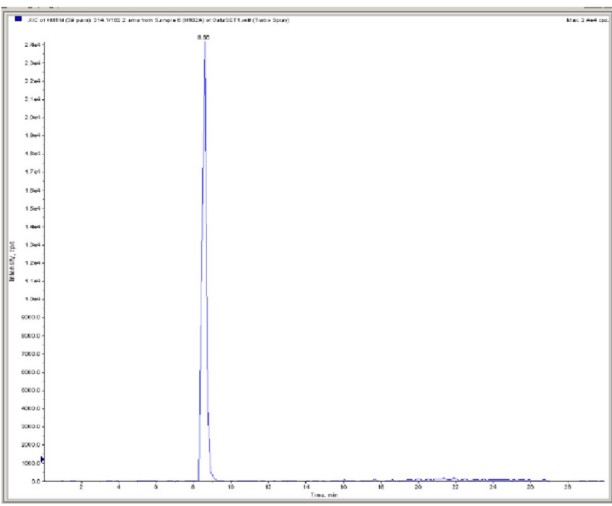

**b**

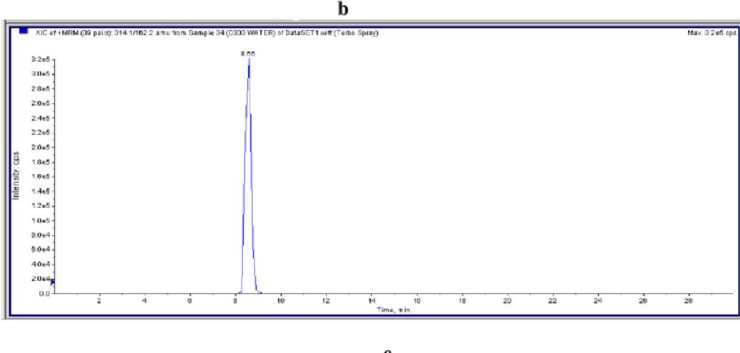

**c**

**Fig 4.** a. Standard chromatogram of Triazophas (standard). b. chromatogram of Triazophas In Blood sample of Case. c. chromatogram of Triazophas in water sample of Case.

samples and Nickel was high in 4 out of 47 urine samples. One of the 36 water samples; seven of the 62 vegetable samples and one sample each of dal and milk had lead levels above the acceptable limit as per FSSAI (Food Safety and Standards Authority of India) guidelines.

## 3.4 Biochemical analysis of samples for mycotoxins

The aflatoxin levels in all the samples of rice and dal were below the critical limits (15 ppb) suggested by FSSAI.

## 3.5 Biochemical analysis of samples for microbiological contamination

About 32% of the drinking water samples were not conforming to the microbiological standards of the Bureau of Indian Standards (BIS). Since the detected microorganisms (E Coli) generally are found in sewage, the presence of these pathogenic bacteria could be due to sewerage water contamination into municipal water pipeline and breach of the sanitation barrier.

From the above results, it was concluded to be a single outbreak with acute neurological symptoms caused due to the presence of Triazophos (Organophosphate) contamination in water at the Household level.

## 4. Discussion

The presence of triazophos in blood, urine and water samples, despite its short half-life, and the time-lapse curve showing characteristic decrease with time, clearly indicate organophosphate contamination. The primary source of pesticide contaminating the drinking water remains unclear however, as per the literature there is a possibility of pesticide leaching into water bodies due to heavy rains and floods [16].

Present study has two limitations—first, it provides evidence of contamination, however, the exact point of contamination couldn't be ascertained. There is a dire need for a more in-depth and integrated study; second, the absence of frank muscarinic and nicotinic clinical picture with only pure CNS findings could be due to low level of pesticide exposure, however, such unique presentations, though rare, have been documented [17].

Though food intoxications are more common, drinking water also carries a potential risk of contaminants, whose sources can be agricultural, industrial or municipal discharges, natural geological formations, run-offs, processes for purification of drinking water, and in distribution channels. Groundwater can also be contaminated with heavy metals. Most common contaminants include lead, aluminium, arsenic, fluoride, pesticides, etc. Heavy metal contamination can occur in a slow, continuous and chronic scale [18,19].

Lead contamination causes effects like decreased intellectual ability, low IQ, anti-social behavior, raised blood pressure, cardiac or renal diseases, infertility and cancers. Flouride contamination may lead to a spectrum of conditions called fluorosis and may also cause osteosarcoma. Aluminium and arsenic may cause peripheral neuropathy, Alzheimer's disease, disorders of reproductive, cardiovascular, immunological, and neurological systems along with carcinomas of Skin, bladder and prostate. Pesticides or their residues can lead to reproductive, immunological, and neurological disorders and even leukemias [19].

Pesticides are used to improve economic potential and control vector-borne diseases. They help in eradication of pests and give high yield, however, they also cause serious negative environmental and health implications. With a threat to environmental, there is a probability of its bioaccumulation, pesticide poisoning leading to foodborne diseases. Organophosphate compounds have largely been used as pesticides in many parts of the world. They are readily available because of insufficient regulations to control their sale [20]. This ease of availability has resulted in a gradual increase in accidental and suicidal poisoning [3]. Globally, each year, poisoning due

to organophosphate compounds are responsible for approximately three million episodes that result in nearly 200,000 deaths. Triazophos is Class Ib (Highly poisonous) pesticide as classified by World Health Organisation [21]. As on 01.01.2021, the formulations with triazophos have been banned for use in India but allowed to be continued to manufacture for export [22].

Pesticide residues are usually not found in human biological samples. In the present study, the mean blood Triazophos pesticide concentrations were: within 24 hrs-17.6 ug/L, 48 hrs-8.0 ug/L and 72 hrs-5.7ug/L. An earlier study [13] on drinking water in urban areas found residues of seven pesticides in borewell water, as well as in the piped municipal water supply. Triazophas was one among the seven with a mean concentration of 0.33 μg/L in the municipal water. Whereas, in the current study, mean concentration of Triazophos was 1.00 μg/L in municipal water of affected households. The concentrations were relatively higher than the earlier study which could have caused the neurological symptoms in the current outbreak. In a study [23] on individuals who consumed pesticides for suicidal reasons, the concentrations of Triazophos was found to be very high: within 24 hr-2948 ug/L, 48 hr-2489 ug/L and 72 hr-1139 ug/L. However, in the present study, the presence of relatively lower concentrations might be due to accidental contamination of pesticides in drinking water. This can be an explanation for the presence of limited CNS symptoms which were not life-threatening.

Organo phosphorous Pesticides has been associated with effects on the function of cholinesterase enzymes, decrease in insulin secretion, disruption of normal cellular metabolism of proteins, carbohydrates and fats, and also with genotoxic effects and effects on mitochondrial function, causing cellular oxidative stress and damage to the nervous and endocrine systems. Community studies have shown association of exposure to organophosphorus pesticides with serious health effects including cardiovascular diseases, negative effects on the male reproductive system and on the nervous system, dementia, and also a possible increased risk for non-Hodgkin's lymphoma [24]. Farm workers being the major users of the pesticides, knowledge related to usage, precautionary measures while handling, spraying and post-usage hygienic practices are essential. Mediti et al (2017) [25] in their study to assess knowledge, practice and self-reported morbidity symptoms of pesticide use among farm women, reported that a majority of them were not aware of toxicity symbols and never read the precautions on the pesticide containers. Inaccessibility was the main reason for the insubstantial use of personal protective equipment (PPEs). Unsafe storage and disposal practices of containers were observed. Weakness (57.3%), headache (52%) and itching of the skin (51.1%) were the common morbidity symptoms up on exposure. A significant association was found between morbidity symptoms and use of PPE and hygiene practices, indicating the importance of such practices.

Though the participants of the present study were not specifically farmers, the outbreak indicates that there is a need to improve awareness of handling pesticides among farmers in the study and prevent any accidental over-usage of pesticides and further drain into the adjoining water bodies or food sources. Similarly, proper washing of vegetables and fruits is to be educated among the residents.

## 5. Conclusions

In conclusion, the probable cause of the outbreak is due to presence of Triazophos (organophosphorous) pesticide contamination in municipal drinking water at household level. The primary source of pesticide contamination into drinking water needs to be identified by an intersectoral approach of government, water works department, public health engineers and other regulatory bodies. Keeping Farmers Health in perspective, Rational usage of pesticides need to be promoted, frequent education and training programs to be conducted to promote awareness and minimize the hazards of pesticide exposure. Diffusion of pesticides into ground

water which were caused by leaching through the soil and unsaturated zone and infiltration through riverbanks and riverbeds needs to be monitored by conducting regular and quarterly surveillance across all districts and states for the presence of residual pesticides in soil, water and food with heightened vigour.

## Supporting information

**S1 File. Clinical.** (This file contains Clinical presentation of Cases and controls from the Hospital and the Community).
(XLSX)

**S2 File. Heavy metals.** (This file contains Heavy metal concentrations in Blood,urine, water and food samples).
(XLSX)

**S3 File. Pesticides.** (This file contains Pesticide concentrations in Blood, Urine and water samples).
(XLSX)

**S4 File. Microbiological.** (This file contains Microbiological analysis of water samples).
(XLSX)

## Acknowledgments

The authors acknowledge Commissioner, Ministry of Health & Family Welfare, Govt. of Andhra Pradesh State, Collector & Joint Collector West Godavari. Dr. Chandra Shekar Reddy, Chairman, Andhra Pradesh Medical Services and Infrastructure Development Corporation (AP MSIDC), Dr. Mohan, Superintendent, Government Hospital, Eluru and special thanks to Mrs. Madhavi, Dy Tahsildar, Govt. of A.P, for their support during the investigation.

The authors are very much grateful to the Indian Council of Medical Research, Ministry of Health & Family Welfare (MoH&FW), Govt. of India, New Delhi and Director, ICMR-NIN for giving an opportunity to conduct the epidemic investigation. The authors acknowledge the services of Dr Dinesh Kumar, Dr. Vakdevi B, NS Kumar, Nasarvali SK and staff for supporting the study.

## Author Contributions

**Conceptualization:** Raghavendra Pandurangi, J. J. Babu Geddam, Sukesh Narayan Sinha.

**Data curation:** J. J. Babu Geddam.

**Formal analysis:** Mahesh Kumar Mummadi, Raghavendra Pandurangi, J. J. Babu Geddam, Sukesh Narayan Sinha, Sivaperumal P., Naveen K. Ramachandrappa, Sree Ramakrishna K., Pagidoju Sreenu.

**Funding acquisition:** J. J. Babu Geddam.

**Investigation:** Mahesh Kumar Mummadi, Raghavendra Pandurangi, J. J. Babu Geddam, Sukesh Narayan Sinha, Ananthan Rajendran, Sivaperumal P., Naveen K. Ramachandrappa, Sree Ramakrishna K., Pagidoju Sreenu.

**Methodology:** Mahesh Kumar Mummadi, Raghavendra Pandurangi, J. J. Babu Geddam, Sukesh Narayan Sinha, Ananthan Rajendran, Naveen K. Ramachandrappa, Sree Ramakrishna K., Pagidoju Sreenu.

**Project administration:** J. J. Babu Geddam.

**Resources:** Mahesh Kumar Mummadi, Raghavendra Pandurangi, J. J. Babu Geddam, Ananthan Rajendran.

**Software:** J. J. Babu Geddam.

**Supervision:** Mahesh Kumar Mummadi, Raghavendra Pandurangi, J. J. Babu Geddam.

**Validation:** J. J. Babu Geddam, Sukesh Narayan Sinha, Ananthan Rajendran, Sivaperumal P.

**Visualization:** J. J. Babu Geddam.

**Writing – original draft:** Mahesh Kumar Mummadi, Raghavendra Pandurangi, J. J. Babu Geddam, Sukesh Narayan Sinha.

**Writing – review & editing:** Mahesh Kumar Mummadi, Raghavendra Pandurangi, J. J. Babu Geddam, Ananthan Rajendran, Sivaperumal P., Naveen K. Ramachandrappa, Sree Ramakrishna K., Pagidoju Sreenu.

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
