## [Decision Letter · Decision Letter 0]

11 May 2021

PONE-D-21-09665

Investigation of an Acute Neurological Outbreak in Eluru, India, 2020

PLOS ONE

Dear Dr. Sinha,

Thank you for submitting your manuscript to PLOS ONE. After careful consideration, we feel that it has merit but does not fully meet PLOS ONE’s publication criteria as it currently stands. Therefore, we invite you to submit a revised version of the manuscript that addresses the points raised during the review process.

We look forward to receiving your revised manuscript.

Kind regards,

Flavio Manoel Rodrigues Da Silva Júnior

Academic Editor

PLOS ONE

3. Thank you for including your ethics statement:  "Ethical Clearance was obtained from the ICMR-NIN Institutional Ethical Committee. IEC Registration Number- ECR/35/Inst/AP/2013, Study Protocol Number-1/I/2021. Informed written consent was taken from the study participants to obtain the samples after explaining the procedures.  ".   

Reviewers' comments:

Reviewer's Responses to Questions

**Comments to the Author**

1. Is the manuscript technically sound, and do the data support the conclusions?

Reviewer #1: Yes

Reviewer #2: No

2. Has the statistical analysis been performed appropriately and rigorously? 

Reviewer #1: Yes

Reviewer #2: Yes

3. Have the authors made all data underlying the findings in their manuscript fully available?

Reviewer #1: No

Reviewer #2: No

4. Is the manuscript presented in an intelligible fashion and written in standard English?

Reviewer #1: Yes

Reviewer #2: No

5. Review Comments to the Author

Reviewer #1: This is a very interesting manuscript about a toxic outcome in India. The manuscript is well-written and the hypothesis of intoxication are well explained. However, some points need to be improved:

1) Biological samples collection and watr collection should be detailed described.

2) Sample preparation for the analysis should be described.

3) Is there any data abour cholinesterase activity of the patients?

4) Is there any limit for triazophos residue in food?

Reviewer #2: In the methods section: it is necessary to better describe the analysis of pesticide detection. I see that it is necessary to put important information as an extractor solvent and which adsorbent used.

In the results section: The presentation of the data is very confusing and would indicate modifying the types of graphs presented. Including that screen prints are rarely used in a scientific article.

In the discussion session: I think it is necessary to place the limitations of the study in the initial part of the discussion.

The article has interesting data, but it needs to be worked on and better addressed in the writing of the article. As it is, I advise you to reject it.

6. PLOS authors have the option to publish the peer review history of their article (what does this mean?). If published, this will include your full peer review and any attached files.

Reviewer #1: No

Reviewer #2: No

---

## [Author Response · Author response to Decision Letter 0]

23 Jun 2021

Respected Reviewers,We are pleased to get an initial remark from you.

Hereby we submit the corrections made accordingly.

Reviewer #1: 

1) Biological samples collection and water collection should be detailed described.

A: As suggested, we have included the methodology of blood, urine and water sample collection.

2) Sample preparation for the analysis should be described.

A: As suggested, we have included sample preparation for the analysis

3) Is there any data about cholinesterase activity of the patients?

A: As we have found organophosphorus in both blood and its metabolites in urine samples of the cases, which was confirmed by high resolution LC-M/S, hence we haven’t analysed samples for any cholinesterase activity.

4) Is there any limit for triazophos residue in food?

A: There is No minimum limit for Triazophos residue in food.

Reviewer #2: 

1) In the methods section: it is necessary to better describe the analysis of pesticide detection. I see that it is necessary to put important information as an extractor solvent and which adsorbent used.

A: As suggested, we have included the methodology of sample preparation with detailed methodology of pesticide analysis

2) In the results section: The presentation of the data is very confusing and would indicate modifying the types of graphs presented. Including that screen prints are rarely used in a scientific article.

A: As suggested, we have included figures and table for better presentation of data. We have removed screen prints and uploaded images in high resolution (PACE approved)

3) In the discussion session: I think it is necessary to place the limitations of the study in the initial part of the discussion.

A: As suggested, we have kept the limitations of the study in earlier paragraph of discussion.

We hope, we have submitted the suggested corrections and ready for more if any. Thank you for consideration.

---

## [Decision Letter · Decision Letter 1]

23 Jul 2021

PONE-D-21-09665R1

Investigation of an Acute Neurological Outbreak in Eluru, India, 2020

PLOS ONE

Dear Dr. Sinha,

Thank you for submitting your manuscript to PLOS ONE. After careful consideration, we feel that it has merit but does not fully meet PLOS ONE’s publication criteria as it currently stands. Therefore, we invite you to submit a revised version of the manuscript that addresses the points raised during the review process.

We look forward to receiving your revised manuscript.

Kind regards,

Flavio Manoel Rodrigues Da Silva Júnior

Academic Editor

PLOS ONE

Reviewers' comments:

Reviewer's Responses to Questions

**Comments to the Author**

1. If the authors have adequately addressed your comments raised in a previous round of review and you feel that this manuscript is now acceptable for publication, you may indicate that here to bypass the “Comments to the Author” section, enter your conflict of interest statement in the “Confidential to Editor” section, and submit your "Accept" recommendation.

Reviewer #1: All comments have been addressed

Reviewer #2: All comments have been addressed

2. Is the manuscript technically sound, and do the data support the conclusions?

Reviewer #1: Yes

Reviewer #2: Partly

3. Has the statistical analysis been performed appropriately and rigorously? 

Reviewer #1: Yes

Reviewer #2: Yes

4. Have the authors made all data underlying the findings in their manuscript fully available?

Reviewer #1: (No Response)

Reviewer #2: Yes

5. Is the manuscript presented in an intelligible fashion and written in standard English?

Reviewer #1: Yes

Reviewer #2: Yes

6. Review Comments to the Author

Reviewer #1: The manuscript was improved after the revision. All my coments were addressed, therefore, the manuscript could be accepted for publication.

Reviewer #2: The abstract is very large and divided into two parts. I suggest that the abstract be reduced to 1 page and the reduction occurs mainly in the introduction to the subject.

Introduction: I found the introduction to this type of article with proposed design unusual. I suggest you improve the introduction by discussing more general outbreak cases, because when treated in this way it looks like an article of regional interest.

Methodology: I am in doubt whether this would actually be an exploratory cross-sectional study or a case-control study. I believe this study is a case-control, where the design is widely used in cases of outbreaks. Most case-control studies use markers to identify the exposure that caused the effect. Please review your study design and if you stick to your decision, I expect plausible arguments for that decision.

Results: It's good

Discussion: I think it is necessary to talk more about toxicological studies related to exposure to a variety of contaminants and the adverse effects that can be observed.

Conclusion: It is not a conclusion of the study to say that this is the first study of the outbreak. The conclusion needs to be made from the main findings found and indicate solutions or possible ways forward.

7. PLOS authors have the option to publish the peer review history of their article (what does this mean?). If published, this will include your full peer review and any attached files.

Reviewer #1: No

Reviewer #2: No

---

## [Author Response · Author response to Decision Letter 1]

19 Sep 2021

Reviewer #1:

Question: The manuscript was improved after the revision. All my comments were addressed; therefore, the manuscript could be accepted for publication.

Answer: Thank you for your kind suggestions.

Reviewer #2:

Question 1: The abstract is very large and divided into two parts. I suggest that the abstract be reduced to 1 page and the reduction occurs mainly in the introduction to the subject.

Answer1: Abstract was reduced to one page as per the suggestions.

Question 2: Introduction: I found the introduction to this type of article with proposed design unusual. I suggest you improve the introduction by discussing more general outbreak cases, because when treated in this way it looks like an article of regional interest.

Answer 2: We have added general outbreak cases in introduction as suggested.

Question 3: Methodology: I am in doubt whether this would actually be an exploratory cross-sectional study or a case-control study. I believe this study is a case-control, where the design is widely used in cases of outbreaks. Most case-control studies use markers to identify the exposure that caused the effect. Please review your study design and if you stick to your decision, I expect plausible arguments for that decision.

Answer 3: After a thorough internal discussion, we have reviewed the study design and appropriately have corrected the design as case-control study. Thank you for the kind suggestion.

Question 4: Results: It's good

Answer 4: Thank you.

Question 5: Discussion: I think it is necessary to talk more about toxicological studies related to exposure to a variety of contaminants and the adverse effects that can be observed.

Answer 5: As suggested, we have discussed few toxicological studies with health effects related to variety of contaminants. 

Question 6: Conclusion: It is not a conclusion of the study to say that this is the first study of the outbreak. The conclusion needs to be made from the main findings found and indicate solutions or possible ways forward.

Answer 6: The conclusion is rewritten by mentioning main findings of the study and related solutions as suggested.

---

## [Editor Report · Decision Letter 2]

15 Oct 2021

Investigation of an Acute Neurological Outbreak in Eluru, India, 2020

PONE-D-21-09665R2

Dear Dr. Sinha,

We’re pleased to inform you that your manuscript has been judged scientifically suitable for publication and will be formally accepted for publication once it meets all outstanding technical requirements.

Kind regards,

Flavio Manoel Rodrigues Da Silva Júnior

Academic Editor

PLOS ONE

---

## [Editor Report · Acceptance letter]

27 Oct 2021

PONE-D-21-09665R2 

Investigation of an Acute Neurological Outbreak in Eluru, India, 2020 

Dear Dr. Sinha:

I'm pleased to inform you that your manuscript has been deemed suitable for publication in PLOS ONE. Congratulations! Your manuscript is now with our production department. 

Kind regards, 

on behalf of

Professor Flavio Manoel Rodrigues Da Silva Júnior 

Academic Editor

PLOS ONE